# LEARNING SAMPLE REWEIGHTING FOR ADVERSARIAL ROBUSTNESS

## ABSTRACT

There has been great interest in enhancing the robustness of neural network classifiers to defend against adversarial perturbations through adversarial training, while balancing the trade-off between robust accuracy and standard accuracy. We propose a novel adversarial training framework that learns to reweight the loss associated with individual training samples based on a notion of class-conditioned margin, with the goal of improving robust generalization. Inspired by MAML-based approaches, we formulate weighted adversarial training as a bilevel optimization problem where the upper-level task corresponds to learning a robust classifier, and the lower-level task corresponds to learning a parametric function that maps from a sample's *multi-class margin* to an importance weight. Extensive experiments demonstrate that our approach improves both clean and robust accuracy compared to related techniques and state-of-the-art baselines.

## 1 INTRODUCTION

While neural networks have been extremely successful in tasks such as image classification and speech recognition, recent work (Szegedy et al., 2014; Goodfellow et al., 2015) has demonstrated that neural network classifiers can be arbitrarily fooled by small, adversarially-chosen perturbations of their input. Notably, Su et al. (2017) demonstrated that neural network classifiers which can correctly classify "clean" images may be vulnerable to *targeted attacks*, e.g., misclassify those same images when only a single pixel is changed.

Recent work has shown a common failing among techniques that uniformly encourage robustness. In particular, there exists an intrinsic tradeoff between robustness and accuracy (Zhang et al., 2019). Bao et al. (2020) investigate this tradeoff from the perspective of classification-callibrated loss theory. Rice et al. (2020) empirically showed that during adversarial training, networks often irreversibly lose robustness after training for a short time. They dubbed this phenomenon *adversarial overfitting* while proposing early stopping as a remedy. The significance of label noise and memorization in the context of adversarial overfitting was demonstrated in Sanyal et al. (2021)—in particular that poor training samples induce fragility to adversarial perturbations due to the tendency of neural networks to interpolate the training data. Methods based on weight and logit smoothing have been proposed as an alternative to early stopping (Chen et al., 2021; Cohen et al., 2019; Salman et al., 2019).

In another line of work, Geometry-Aware Instance Reweighted Adversarial Training (GAIRAT; Zhang et al. (2021)), Weighted Margin-aware Minimax Risk (WMMR; Zeng et al. (2021)), and Margin-Aware Instance reweighting Learning (MAIL; Wang et al. (2021)) were developed to address adversarial overfitting by controlling the influence of training examples via importance / loss weighting. Intuitively, the samples assigned a low weight correspond to samples on which the classifier is already sufficiently robust. However, existing methods rely on rigid approximations of the margin and employ heuristic weighting schemes that rely on careful choices of hyperparameters.

This work builds upon these observations. We present BiLAW (Bilevel Learnable Adversarial reWeighting), an approach that explicitly learns a parametric function (e.g. represented by a small feed-forward network) that assigns weights to the loss suffered by a classifier, associated with individual training samples. The sample weights are learned as a function of the classifier *multiclass margins* of the sample, according to the weights' effect on *robust generalization*. We employ a bi-level optimization formulation Bracken & McGill (1973) and leverage a validation set, where the *upper-level* objective corresponds to learning the parameters of a robust classifier, while the

*lower-level* objective corresponds to learning a function that predicts sample weights that improve improve robustness on a validation set. Our approach alternates between iteratively updating the parametric sample weights and updating the classifier network parameters.

**Contributions** We note that while sample weighting (Zhang et al., 2021; Yi et al., 2021; Wang et al., 2021) has been investigated in the context of robust training, as far as we know this is the first work to explore a method to use a validation set to learn weights to induce robust generalization. Our contributions consist of the following:

1. Our approach performs sample re-weighting during training. We propose a bilevel optimization formulation to learn a mapping from multi-class margins to weights according to the robust loss suffered by a classifier on a validation set.

2. We show that when our weighting function corresponds to a neural network, the magnitude of a sample's weight directly corresponds to the vulnerability of the classifier at that sample.

3. We evaluate the practical performance of BiLAW on MNIST (LeCun & Cortes, 2010), F-MNIST (Xiao et al., 2017), CIFAR-10, and CIFAR-100. (Krizhevsky et al.) and demonstrate that it improves clean accuracy up to $6\%$ and robust test accuracy by up to $5\%$ compared to TRADES and other state-of-the-art sample reweighting methods on CIFAR-10.

This paper is organized as follows. In Section 2, we review the notation and background of cost-aware and robust classification. In Section 3 we describe our method in full, including the sample weighting method. In Section 4, we provide ablative experiments and demonstrate the efficacy of our framework by comparing clean and robust test performance on MNIST, F-MNIST, CIFAR-10, and CIFAR-100 to adversarial training and two recent, state-of-the-art sample weighting methods.

## 2   PRELIMINARIES AND RELATED WORK

In this section, we briefly present background terminology pertaining to adversarially robust classification, sample reweighting and bilevel optimization.

**Notations**   An ReLU network is a neural network such that all nonlinear activations are ReLU functions, where we denote the ReLU activation by $\sigma : \mathbb{R} \to \mathbb{R}$, $\sigma(x) = \max\{0, x\}$. Informally, we define $\sigma : \mathbb{R}^d \to \mathbb{R}^d$ by $\sigma(x) = [\sigma(x_1), \ldots, \sigma(x_d)]$. Let $f_\theta : \mathbb{R}^d \to [0, 1]^k$ be a feedforward ReLU network with $l$ hidden layers and weights $\theta$; for example, $f$ may map from a $d$-dimensional image to a $k$-dimensional vector corresponding to likelihoods for $k$ classes.

Given a training set of $m$ sample-label pairs $(x_i, y_i)$ drawn from a training data distribution $\mathcal{D}$, we associate a *weight* $w_i$ with each training sample. Informally, these weights characterize the effect of the sample on the generalization of the network (i.e. samples with large weights promote robust generalization and visa versa). Given a loss function $\ell : \mathbb{R}^k \times \mathbb{R}^k \to \mathbb{R}$, we denote the empirical *weighted* training loss suffered by a network with parameters $\theta$ on $m$ training samples with weights $w$ to be $\mathcal{L}_{\text{tr}}(\theta, w) = \sum_{i=1}^{m} w_i \ell(y_i, f_\theta(x_i))$ such that $w_i \geq 0$ and $\sum_i w_i = 1$. For brevity, we write $\ell_i(\theta) = \ell(y_i, f_\theta(x_i))$. Additionally, if $w$ is left unspecified, $\mathcal{L}$ corresponds to the unweighted mean over empirical losses. Likewise, the *unweighted* validation loss of $n$ samples is denoted $\mathcal{L}_{\text{val}}(\theta)$.

### 2.1   ROBUST CLASSIFICATION AND ADVERSARIAL OVERFITTING

Consider the network $f_\theta : \mathbb{R}^d \to \mathbb{R}^k$, where the input is $d$-dimensional and the output is a $k$-dimensional vector of likelihoods, with $j$-th entry corresponding to the likelihood the image belongs to the $j$-th class. The associated classification is then $c(x) = \arg\max_{j \in [1,k]} f_{\theta,j}(x)$. In adversarial machine learning, we are not just concerned that the classification be correct, but we also want to be robust against adversarial examples, i.e. small perturbations to the input which may change the classification to an incorrect class. We define the notion of $\epsilon$-robustness below:

**Definition 1 ($\epsilon$-robust)**  *$f_\theta$ is called $\epsilon$-robust with respect to norm $p$ at $x$ if the classification is consistent for a small ball of radius $\epsilon$ around $x$:*

$$c(x + \delta) = c(x), \forall \delta : ||\delta||_p \leq \epsilon. \tag{1}$$

Note that the $\epsilon$-robustness of $f_\theta$ at $x$ is intimately related to the uniform and local Lipschitz smoothness of $f_\theta$ around $x$. Recall that a function $f$ has finite, global Lipschitz constant $L > 0$ with respect to norm $|| \cdot ||$, if

$$\exists L \geq 0 \text{ s.t. } |f(x) - f(x')| \leq L \cdot ||x - x'||, \forall x, x' \in X. \tag{2}$$

An immediate consequence of Eq. 1 and Eq. 2 is that if $f_\theta$ is uniformly $L$-Lipschitz, then $f_\theta$ is $\epsilon$-robust at $x$ with $\epsilon = \frac{1}{2L}(P_a - P_b)$ where $P_a$ is the likelihood of the most likely outcome, and $P_b$ is the likelihood of the second most likely outcome (Salman et al., 2019). The piecewise linearity of ReLU networks facilitates the extension of this consequence to the *locally Lipschitz* regime. $L$ corresponds to the norm of the affine map characterized by $f$ conditioned on input $x$. These properties were previously used to characterize the robustness of a network at a sample (and the weight associated with the sample) (Zeng et al., 2021; Yi et al., 2021; Wang et al., 2021).

The minimal $\ell_p$-norm perturbation $\delta_p^*$ required to switch an sample's label is given by the solution to the following optimization problem:

$$\delta_p^* = \arg\min ||\delta||_p \quad \text{s.t.} \quad c(x) \neq c(x + \delta).$$

A significant amount of existing work relies on a first-order approximations and Hölder's inequality to recover $\delta^*$, justifying the popularity of inducing robustness by controlling global and local Lipschitz constants. More concretely, given a $\ell_p$ norm and radius $\epsilon$, a typical goal of robust machine learning is to learn classifiers that minimize the *robust loss* on a training dataset:

$$\min_\theta \mathbb{E}_{(x,y) \sim \mathcal{D}} \left[ \max_{||\delta||_p \leq \epsilon} \ell(y, f_\theta(x + \delta)) \right].$$

For brevity we will denote the robust analogue of a loss $\mathcal{L}$ as $\hat{\mathcal{L}}$, indicating this is the robust counterpart of $\mathcal{L}$, differentiated by the "inner" maximization problem.

## 2.2 MARGIN-AWARE REWEIGHTING

In the framework of cost-sensitive learning, weights are assigned to the loss associated with individual samples. The goal of cost-sensitive learning is to minimize the empirical *weighted training loss*:

$$\min_\theta \mathcal{L}_{\text{tr}}(\theta, w) := \sum_{i=1}^m w_i \ell_i(\theta).$$

Previous work in cost-sensitive learning for adversarial robustness (Zhang et al., 2020; 2021; Zeng et al., 2021) typically substitutes the robust loss $\hat{\mathcal{L}}_{\text{tr}}$ for $\mathcal{L}_{\text{tr}}$ and largely focuses on designing heuristic functions of various notions of margin to use for the sample weight $w_i$.

For example, in GAIRAT (Zhang et al., 2021; 2020), the margin is defined as the least number of PGD steps, denoted $\kappa$, that leads the classifier to make an incorrect prediction. The sample's weight is computed as $\omega_{\text{GAIRAT}}(x_i) = \frac{1}{2}(1 + \tanh(\lambda + 5(1 - 2\kappa/K)))$ with hyperparameters $K$ and $\lambda$. A small $\kappa$ indicates that the sample lies close to the decision boundary. Larger $\kappa$ values imply that associated samples lie far from the decision boundary, and are therefore more robust, requiring smaller weights. However, due to the non-linearity of the loss-surface in practice, PGD-based attacks with finite iterations may suffer from the same issues that plague standard iterative first-order methods in non-convex settings. In other words, $\kappa$ is heavily dependent on the optimization path taken by PGD. This is demonstrated by GAIRAT's vulnerability on sophisticated attacks, e.g. AutoAttack (Croce & Hein, 2020b).

Zhang et al. (2020) define the margin as the difference between the loss of a network suffered at a clean sample and its adversarial variant. Zeng et al. (2021); Wang et al. (2021) propose a definition of margin corresponding to taking differences between logits, as follows.

**Definition 2 ((Zeng et al., 2021; Wang et al., 2021))** *The margin of a classifier $f_\theta$ on sample $(x_i, y_i)$ is the difference between the classifier's confidence in the correct label $y_i$ and the maximal probability of an incorrect label $t$, $margin(f_\theta, x_i, y_i) = p(f(x_i) = y_i) - max_{t \neq y_i} p(f_\theta(x_i) = t)$.*

Given this definition, Zeng et al. (2021); Wang et al. (2021) propose to use an exponential (WMMR) and sigmoidal (MAIL) functions respectively: $\omega_{\text{WMMR}}(x_i) = \exp(-\alpha m)$ with parameter $\alpha$, and

$\omega_{\text{MAIL}}(x_i) = \text{sigmoid}(-\gamma(m - \beta))$ with parameters $\gamma$ and $\beta$. WMMR and MAIL rely on the local linearity of ReLU networks and the fact that for samples near the margin, the relative scale of predicted class-likelihoods directly corresponds to the distance to the decision boundary. However, similarly to GAIRAT's $\kappa$, even for samples very close to the decision boundary, simple functions of the difference between class likelihoods may not necessarily correspond to the true distance to the decision boundary. Here we propose a more fine-grained notion of margin, the *multi-class margin*, and design a method to learn a mapping between the margin at a sample and its associated weight.

Previous work has explored theoretical notions of a *multi-class margin*. For example, Zou (2005) defined the *margin vector* in the context of boosting as a proxy for a vector of conditional class probabilities. However, this notion of margin is unaware of the true class of a sample. In contrast, the multi-class margin proposed by Saberian & Vasconcelos (2019); Cortes et al. (2013) are both closely related to Wang et al. (2021); Zeng et al. (2021), i.e. defined as the minimal distance between an arbitrary predicted logit and the logit of the true class.

In Fig. 1 we explore the relationship between the logits of a network evaluated at a clean sample and the predicted class of the adversarially perturbed variant. Methods which rely on the canonical notions of margin reasonably assume that samples at which a classifier is vulnerable have small margin according to Def. 2, i.e. the magnitude of the smallest difference between the logits of any class and the logit corresponding the true class is small. However, we demonstrate in Fig. 1(b) for both vulnerable and robust classifiers that while the majority of predictions made by networks on adversarial samples do correspond to the classes with minimal margin, a significant number do not. In other words, the class for which the margin is smallest does not always correspond to the adversarial class. Furthermore, this issue is exacerbated for robust networks as shown by the difference in count distribution between networks whose relative robustness varies.

## 2.3 BI-LEVEL OPTIMIZATION AND META-LEARNING

Bilevel optimization, first introduced by Bracken & McGill (1973) is a framework for mathematical optimization involving nested optimization problems. A typical bilevel optimization problem takes on the following form:

$$\min_{x \in \mathbb{R}^p} \Phi(x) := f(x, y^*(x)) \quad \text{s.t.} \quad y^* \in \arg\min_{y \in \mathbb{R}^p} g(x, y), \tag{3}$$

where $f$ and $g$ are respectively denoted the *upper-level* and *lower-level* objectives. The goal of the framework is to minimize the primary objective $\Phi(x)$ with respect to $x$ where $y^*(x)$ is obtained by solving the lower-level minimization problem. The framework of bilevel optimization has seen wide adoption by the machine learning community—in particular in the context of hyperparameter tuning (Jenni & Favaro, 2018; Ren et al., 2018) and meta-learning (Finn et al., 2017; Rajeswaran et al., 2019). Our proposed algorithm has some similarity to the meta-learning literature (Finn et al., 2017; Rusu et al., 2019; Rajeswaran et al., 2019; Eshratifar et al., 2018). Most notably, the Model-Agnostic Meta-Learning MAML) algorithm by Finn et al. (2017) incorporates gradient information for the meta-learning setting. The application of meta-learning as an instance of bilevel optimization has been explored in the context of sample reweighting. In particular, Ren et al. (2018); Jenni & Favaro (2018); Shu et al. (2019) proposed methods to address learning with noisy labels by reweighting the gradients associated with the losses at individual samples based on balancing performance on a curated validation set and the corrupted training set.

## 3 BiLAW: LEARNING SAMPLES WEIGHTS FOR ADVERSARIAL TRAINING

In this section, we propose BiLAW, a new learning framework for robust training. There are 2 main novelties in our new learning scheme compared to existing robust training methods: First, we consider a more reasonable assumption leveraging the concept of *multi-class margin* in robust training, where good weights should be aware of both the margin associated with each class, as well as the true class associated with the sample. Second, as opposed to related work which defines an explicit formula for the weights dependent on the margin, we propose to *learn* the weights as part of training the classification model. Specifically, we define the weights as a function of a multi-class margin, and parameterize this function using a small auxiliary network. We formulate this as a bi-level optimization problem and learn the weights iteratively alongside the neural network parameters.

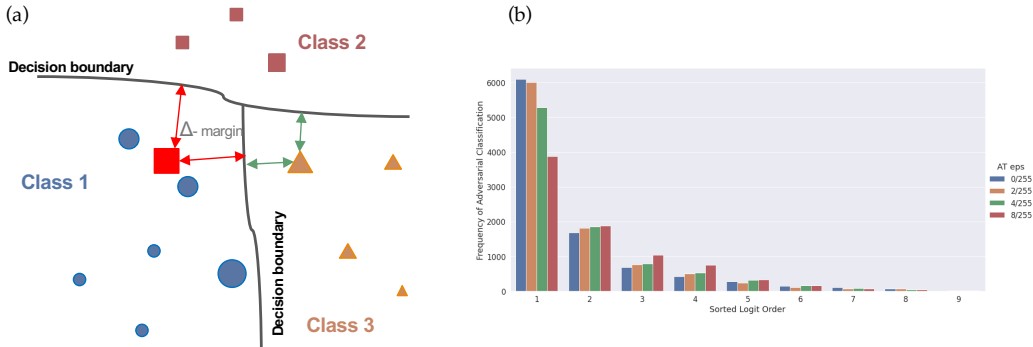

Figure 1: (**a**) Diagram of multiclass margin. Larger samples denote samples that should be assigned large weight, e.g., are misclassified or close to the decision boundary. Green (red) arrows denote entries in the multiclass margin vector for a correctly (incorrectly) classified sampled. (**b**) Sorted logit order and frequency of adversarial classification. Number of instances where the prediction of an adversarial sample corresponds to its $i$-th largest logit in CIFAR-10 (ignoring the $0$-th logit/samples for which the prediction does not change). Colors represent the perturbation budget used during adversarial training (i.e. varying degrees of robustness). Perturbations are computed using $\ell_\infty$-PGD with 10 iterations and a maximum budget of $0.031$.

### 3.1 MULTI-CLASS MARGIN REWEIGHTING

We extend the logit-based definitions of margin applied in Zeng et al. (2021); Wang et al. (2021) and define the multi-class margin of a classifier at a sample as follows.

**Definition 3** *The multi-class margin of a classifier $f_\theta$ on sample $(x_i, y_i)$, denoted $\Delta : [0,1]^k \to [-1,1]^k$, is a $k$-dimensional vector whose $j$-th entry, $\Delta^{(j)}(f_\theta(x_i), y_i)$, is the difference between the classifier's confidence in the correct label $y_i$ and the classifier's confidence in label $j$, $\Delta^{(j)}(f_\theta(x_i), y_i) = p(f_\theta(x_i) = y_i) - p(f_\theta(x_i) = j)$.*

For brevity we denote $\Delta(f_\theta(x_i), y_i)$ as $\Delta_i$. Note that the multi-class margin exhibits two qualities:

1. Correct/incorrect classification is implicit in the definition, as negative values indicate an incorrect classification.
2. The true class of the sample is also implicit—i.e. the index with element zero (assuming the sample does not lie exactly on a decision boundary separating the true class from another).

In particular, we highlight the second quality. Prior work has demonstrated that the distribution of predictions made on adversarial samples is not necessarily uniform over all classes (Abbasi & Gagné, 2017). In other words, vulnerable samples and their associated adversarial perturbations may concentrate about certain classes more than others. We demonstrate in the results that networks exhibit non-uniform robustness per-class.

To learn the sample weights as a function of the multiclass margin, we construct an auxiliary neural network with a single hidden layer, whose parameters are denoted $\mu$ and whose inputs are the multi-class margins. The weight of the $i$-th training sample is then computed as $w_i = \omega_\mu(\Delta_i)$. In general, we denote the function used to map from margin to weight $\omega(\cdot)$. A question that arises is what loss function should be used to train this auxiliary network. We design a bilevel optimization approach leveraging the validation set to learn the auxiliary network parameters $\mu$.

### 3.2 BILEVEL OPTIMIZATION

We exploit a validation set to jointly learn a parametric weighting function $\omega_\mu$ on the training samples and a classifier which jointly minimize the associated weighted robust error. Let $\hat{\mathcal{L}}_{\text{tr}}(\theta_t, w_t) = \sum_{i=1}^m w_{t,i}\hat{\ell}_i(\theta_t)$, where $w_{t,i} = \omega_{\mu_t}(\Delta_i)$ be the *weighted* robust training loss with respect to parameters $\theta_t$ and $\mu_t$ at time $t$. Additionally, $w_{t,i} \geq 0$ and $\sum_{i=1}^{m_b} w_{t,y} = 1$. Intuitively, the

samples with high weights should improve robust generalization—this is quantified by the robust error evaluated on a held-out validation set. Let $\hat{\mathcal{L}}_{\text{val}}(\theta_t) = \frac{1}{n} \sum_{i=1}^{n} \hat{\ell}_i(\theta_t)$ be the *unweighted* robust validation loss associated with $\theta_t$. Following the meta-learning principle, we seek weights such that the minimizer of the weighted robust training loss maximizes robust accuracy on the unweighted validation set—i.e. solve the following bilevel optimization problem:

$$\arg\min_{\theta} \hat{\mathcal{L}}_{\text{tr}}(\theta, \omega_{\mu^*}(\Delta)) \quad \text{s.t.} \quad \mu^* \in \arg\min_{\mu} \hat{\mathcal{L}}_{\text{val}}(\theta) \tag{4}$$

The workflow of the reweighting procedure is illustrated in Fig. 2(A) and the reweighting algorithm is presented in Alg. 1. We provide a high-level overview of the procedure below.

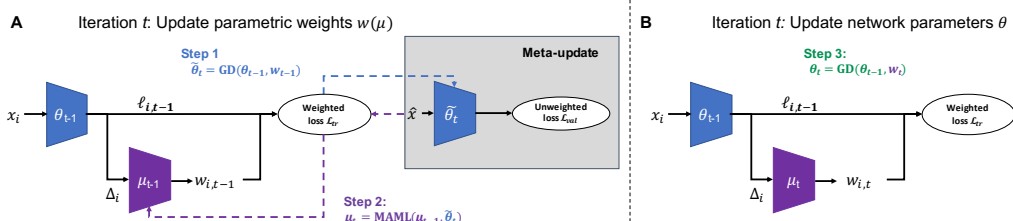

Figure 2: **BiLAW** Framework. (**A**) MAML-inspired sample weighting. Step 1: intermediate parameters $\tilde{\theta}_t$ are computed by pseudo-update of $\theta_t$. Step 2: Validation loss gradients (calculated via back-propagation through the weighted training loss) are used to update the parameters of the auxiliary weighting network $\mu_t$. (**B**) Step 3: network parameters $\theta_t$ updated using new weights $w_t$.

Our approach is composed of three steps. Steps 1 and 2 rely on the MAML-trick (Finn et al., 2017), which substitutes one-step updates $\mu_t$ for $\mu^*$ and iteratively solves the upper-level problem. In this context, $\mu_t$ is updated according to the gradient of the unweighted robust validation loss with respect to the sample weights. We note that this method necessitates computation of a *pseudo-update* in order to compute this gradient:

**Step 1** Pseudo update of classifier parameters $\tilde{\theta}_t$ (Step 1 in Fig. 2, line 5 in Alg. 1)

$$\tilde{\theta}_t = \theta_t - \beta \nabla_\theta \hat{\mathcal{L}}_{\text{tr}}(\theta_t, w_{t-1}) \tag{5}$$

The pseudo parameters $\tilde{\theta}_t$ are then used as a surrogate for $\theta_t$ in optimizing for $\mu$:

**Step 2** Update parameters $\mu_t$ of the auxiliary network (Step 2 in Fig. 2, line 6 in Alg. 1)

$$\mu_t = \mu_{t-1} - \frac{\alpha\beta}{mn} \sum_{j=1}^{m} \left( \sum_{i=1}^{n} \left( \left. \frac{\partial \hat{\ell}_i^{\text{val}}(\tilde{\theta})}{\partial \tilde{\theta}} \right|_{\tilde{\theta}_t} \right)^\top \left. \frac{\partial \hat{\ell}_j^{\text{tr}}(\theta)}{\partial \theta} \right|_{\theta_{t-1}} \right) \left. \frac{\partial w}{\partial \mu} \right|_{\mu_t}, \tag{6}$$

where $\alpha$ and $\beta$ are the step size used in the pseudo and auxiliary network updates, respectively.

**Step 3** Update parameters of classifier network (Step 3 in Fig. 2, line 8 in Alg. 1)

$$\theta_{t+1} = \theta_t - \beta \nabla_\theta \hat{\mathcal{L}}_{\text{tr}}(\theta_t, w_t) \tag{7}$$

One interpretation of this procedure is that we take a pseudo-step using $\theta_{t-1}$ and $\mu_{t-1}$ (Step 1), calculate the best auxiliary network parameters $\mu_t$ in *hindsight* that improve generalization, by minimizing the validation loss with $\tilde{\theta}_t$, (Step 2), and then derive the "real" update for $\theta_{t-1}$ by minimizing the weighted training loss using the new weights $\mu_t$ (Step 3). The detailed derivation of the gradient update is provided in the appendix. Note that the term $\frac{1}{n} \sum_{i=1}^{n} \left( \left. \frac{\partial \hat{\ell}_i^{\text{val}}(\tilde{\theta})}{\partial \theta} \right|_{\tilde{\theta}_t} \right)^\top \left. \frac{\partial \hat{\ell}_j^{\text{tr}}(\theta)}{\partial \theta} \right|_{\theta_{t-1}}$

in Eq. (6) represents the correlation between the gradient of the $j$-th training sample computed on the training loss and the average gradient of the validation data calculated on the robust validation loss. As a consequence, if the gradient of the loss with respect to the network parameters at time $t$ for training sample $j$ is aligned with the average gradient of the meta-loss, it will be considered a beneficial sample for generalization and its weight will be increased. Conversely, the weight of the sample is suppressed if the gradient is anticorrelated with the average validation set-gradient.

---

**Algorithm 1 BiLAW** training procedure

---

**Input:** Training data $\mathcal{D}$, validation-data set $\hat{\mathcal{D}}$, max iterations $T$, learning rates $\alpha, \beta$
**Output:** Classifier parameters $\theta$

1:   $t \leftarrow 0$
2:   Initialize $\theta_0, \mu_0, w_0 = \omega_{\mu_0}(\Delta)$
3:   **for** $t \leq T$ **do**
4:      $(X, y) \sim \mathcal{D}, (\hat{X}, \hat{y}) \sim \hat{\mathcal{D}}$
5:      $\tilde{\theta}_t \leftarrow \theta_t - \beta \cdot \nabla_\theta \hat{\mathcal{L}}_{\text{tr}}|_{\tilde{\theta}_t, w_t}$
6:      $\mu_{t+1} = \mu_t - \alpha \nabla_\mu \hat{\mathcal{L}}_{\text{val}}|_{\theta_t, w_t}$          $\triangleright$ compute $\nabla_\mu \hat{\mathcal{L}}_{\text{val}}|_{\tilde{\theta}_t, w_t}$ via backpropagation according to Eq. 6
7:      compute $w_{t+1} = \omega_{\mu_{t+1}}(\Delta)$               $\triangleright$ compute $\Delta$ with respect to $\theta_t$ according to Def 3
8:      $\theta_{t+1} = \theta_t - \beta \nabla_\theta \hat{\mathcal{L}}_{\text{tr}}|_{\theta_t, w_{t+1}}$
9:   **end for**
10: **return** $\theta_T$

---

## 4   EXPERIMENTS

In this section, we evaluate the efficacy of our framework on a variety of datasets, and demonstrate that our technique improves robustness while preserving clean accuracy. We introduce three variants based on our reweighting technique:

1) Non-parametric reweighting: we learn weights using the *weighted* adversarial cross-entropy loss where the weight $w_{j,t}$ for sample $j$ at iteration $t$ is proportional to the correlation between the training loss gradient and the average validation loss gradient: $\frac{1}{n} \sum_{i=1}^n \left( \frac{\partial \hat{\ell}_i^{\text{val}}(\tilde{\theta})}{\partial \theta} \Big|_{\tilde{\theta}_t} \right)^\top \frac{\partial \hat{\ell}_j^{\text{tr}}(\theta)}{\partial \theta} \Big|_{\theta_{t-1}}$.

2) BiLAW (Parametric reweighting, Sec. 3) trained using the *weighted* adversarial cross-entropy loss.

3) BiLAW can be modified to BiLAW-TRADES: Parametric reweighting trained with the TRADES loss (Zhang et al., 2019), i.e. we solve

$$\min_\theta \sum_i \ell(f_\theta(x_i), y_i) + 1/\lambda(w_i \cdot \text{KL}(f_\theta(x_i), f_\theta(x_i + \delta))), \tag{8}$$

where $\ell()$ corresponds to the standard cross-entropy loss, KL corresponds to the KL-divergence, $\delta$ corresponds to an adversarially perturbation, and $w_i = \omega_\mu(\Delta_i)$: the parametric map applied to the multi-class margin of $f_\theta$ at $x_i$. For all experiments, we set $1/\lambda = 6$, and define $\omega$ to be a single hidden-layer fully connected ReLU network with 128 hidden units and a sigmoid activation. Furthermore, to enforce aforementioned constraints, we normalize the weights per-batch—i.e. $w_i = w_i / \sum_j w_j$.

### 4.1   PERFORMANCE EVALUATION

We evaluate the performance of our approach compared to plain training, adversarial training (AT) (Madry et al., 2018), GAIRAT (Zhang et al., 2021), WMMR (Zeng et al., 2021), and MAIL (Wang et al., 2021). All experiments are run on a single RTX 2080 Ti. When applying our approach and variants, two validation sets of size 1000 are extracted from the training set: one is used to learn the auxiliary network parameters, and the second is used for early stopping. This results in a smaller training set for BiLAW, while the training sets of competing methods are unaltered.

In Table 1, we evaluate BiLAW using two relatively small networks on two datasets: MNIST (LeCun & Cortes, 2010) and Fashion MNIST (Xiao et al., 2017). Tiny-CNN is a convolutional network with 2 convolutional and 2 dense layers. FC1 corresponds to a single hidden layer feedforward network with 1024 hidden units. The details of the architectures are given in the Appendix. We consider robustness with respect to $\ell_\infty$ distance. We use three criteria: clean test accuracy (clean), robust test accuracy (PGD) for a given threshold $\epsilon$ and AutoAttack (AA). Robust test accuracy is computed using Projected Gradient Descent (PGD) (Madry et al., 2018) with 20 iterations. In all testcases, our method matches the performance of GAIRAT and out-performs the other methods for clean and PGD accuracy and we out-perform all reweighting methods on AA accuracy. However, we note the overall distribution of both clean and robust accuracy is tight. We note a potential drawback of reweighting

algorithms: the MNIST and F-MNIST datasets contain a non-trivial number of misclassified samples which can influence performance (Müller & Markert, 2019). For algorithms which perform weighted training, possible large weights on outliers or mislabeled examples may influence classification performance. We will investigate this in the context of adversarial training in future work.

Table 1: MNIST/F-MNIST comparison for plain, AT, GAIRAT, WMMR ($\alpha_{\text{train}} = 0.1$, $\alpha_{\text{test}} = 2$), MAIL ($\gamma = 5$, $\beta = 0.05$) and **BiLAW** using standard robust loss. **Best** result is underlined and bolded and **second best** is bolded.

| | Tiny-CNN | | | | | | FC1 | | | | | |
|---|---|---|---|---|---|---|---|---|---|---|---|---|
| | perturbation: $\ell_\infty$ | | | perturbation: $\ell_2$ | | | perturbation: $\ell_\infty$ | | | perturbation: $\ell_2$ | | |
| | Clean | PGD | AA | Clean | PGD | AA | Clean | PGD | AA | Clean | PGD | AA |
| *MNIST* | $\epsilon = 0.1$ | | | $\epsilon = 0.3$ | | | $\epsilon = 0.1$ | | | $\epsilon = 0.3$ | | |
| plain | **99.1** | 21.7 | 9.1 | **99.2** | 96.9 | 36.4 | 98.4 | 1.7 | 0.0 | 98.3 | 90.3 | 16.1 |
| AT | 99.0 | **95.9** | 93.7 | **99.1** | 98.2 | **96.1** | 98.4 | 92.9 | 90.4 | 98.4 | **97.4** | 95.3 |
| GAIRAT | **99.1** | 96.7 | 91.1 | **99.2** | **98.8** | 90.3 | **99.0** | 93.2 | 89.7 | 98.8 | 97.6 | 89.2 |
| WMMR | 98.8 | 94.3 | 90.2 | 99.0 | 98.5 | 91.7 | 98.9 | 92.8 | 89.4 | 98.2 | 97.2 | 89.8 |
| MAIL | 98.6 | 95.1 | 91.4 | 98.7 | 98.6 | 95.4 | 98.4 | **93.1** | **91.3** | 98.1 | **97.4** | 94.2 |
| BiLAW(ours) | **99.2** | 96.7 | **91.7** | **99.2** | 98.9 | 95.4 | **99.1** | 93.1 | **91.6** | 98.6 | 97.6 | 94.4 |
| *F-MNIST* | $\epsilon = 0.1$ | | | $\epsilon = 0.3$ | | | $\epsilon = 0.1$ | | | $\epsilon = 0.3$ | | |
| plain | **89.6** | 1.5 | 0.0 | 89.7 | 42.9 | 0.0 | **98.5** | 0.0 | 0.0 | 89.3 | 57.2 | 0.0 |
| AT | 86.4 | 70.1 | 68.3 | 91.9 | 79.6 | 77.9 | 87.0 | 68.7 | 66.3 | 91.1 | 80.1 | **76.0** |
| GAIRAT | 86.4 | 77.6 | 64.3 | **92.3** | **81.1** | 70.3 | 87.1 | **70.2** | 61.4 | 91.1 | 81.0 | 70.4 |
| WMMR | 86.2 | 77.3 | 64.1 | 92.1 | 80.6 | 71.4 | 86.9 | 68.4 | 61.3 | 91.1 | 78.4 | 70.9 |
| MAIL | 86.4 | 76.9 | **68.6** | 92.2 | 80.5 | 76.2 | 90.1 | 69.3 | **66.4** | 90.6 | 79.3 | 75.9 |
| BiLAW(ours) | **86.6** | **77.4** | 68.8 | **92.4** | 81.3 | **76.6** | 87.3 | 70.6 | 66.7 | **91.4** | 80.9 | 76.1 |

Table 2: CIFAR-10 comparison for AT, GAIRAT, WMMR, non-parametric weighting, and BiLAW with standard adversarial training (BiLAW) and with TRADES loss (BiLAW-TRADES). We perform AA on 1000 samples in the test set. **Best** result is underlined and bolded and **second best** is bolded.

| | Small-CNN | | | | | | WRN-10-32 | | | | | |
|---|---|---|---|---|---|---|---|---|---|---|---|---|
| | perturbation: $\ell_\infty$ | | | perturbation: $\ell_\infty$ | | | perturbation: $\ell_\infty$ | | | perturbation: $\ell_\infty$ | | |
| | Clean | PGD | AA | Clean | PGD | AA | Clean | PGD | AA | Clean | PGD | AA |
| *CIFAR-10* | $\epsilon = 0.0078$ | | | $\epsilon = 0.031$ | | | $\epsilon = 0.0078$ | | | $\epsilon = 0.031$ | | |
| GAIRAT | 79.0 | 54.7 | 48.1 | 79.0 | **55.6** | 40.7 | **86.4** | 73.6 | 63.1 | 84.7 | 56.8 | 43.4 |
| WMMR | 78.7 | 58.9 | 51.2 | **81.7** | 49.1 | 39.1 | 85.9 | 70.9 | 67.4 | 80.6 | 49.5 | 40.6 |
| MAIL | 76.8 | **64.3** | **59.2** | **81.9** | 53.3 | 40.6 | 84.3 | 74.1 | **73.7** | 83.2 | 53.7 | **52.0** |
| AT | 78.7 | 58.7 | 56.6 | 79.6 | 45.6 | 42.9 | 85.9 | 71.3 | 69.5 | 85.9 | 52.0 | 48.0 |
| TRADES ($1/\lambda = 6$) | **79.2** | 58.9 | 56.8 | 78.9 | 54.8 | **51.7** | 84.6 | 73.9 | 73.1 | 83.1 | 53.9 | **52.1** |
| Non-parametric weighting | 79.7 | 60.0 | 47.3 | 81.3 | 52.2 | 40.6 | **86.4** | 73.7 | 62.3 | **86.6** | 52.8 | 42.9 |
| BiLAW (ours) | 79.7 | 63.6 | 56.7 | 80.4 | 55.4 | 45.3 | 87.1 | 74.2 | 71.3 | **87.4** | **58.6** | 51.4 |
| BiLAW-TRADES (ours) | 79.1 | **64.8** | 61.5 | 80.2 | **56.2** | 52.6 | 86.2 | **74.8** | 74.2 | 86.1 | **58.9** | 53.6 |

In Table 2, we evaluate our method using the two architectures used in Zhang et al. (2021) on CIFAR-10 (Krizhevsky et al.): a 6-layer convolutional network and a Wide-Resnet-10-32 (WRN-32-10) (Zagoruyko & Komodakis, 2016), with details provided in the Appendix. We run each method for 100 epochs with training and validation batch sizes set to 128 using SGD + momentum. A standard learning rate schedule is implemented with the initial learning rate of 0.1 divided by 10 at Epoch 30 and 60, respectively. **BiLAW** strictly outperforms AT with respect to both clean and robust accuracy and generally outperforms GAIRAT and WMMR with respect to clean and robust accuracy on CIFAR-10 (up to 10%). In particular, BiLAW consistently achieves superior clean test accuracy in all testcases, except for the $\ell_\infty$ small-CNN ($\epsilon = 0.031$). On the WRN $\ell_\infty$

Table 3: CIFAR-100 comparison for AT, GAIRAT, WMMR, and BiLAW with the TRADES loss. We perform AA on 1000 samples in the test set. The **best** result is underlined and bolded and the **second best** is bolded.

| | WRN-32-10 | | |
|---|---|---|---|
| | perturbation: $\ell_\infty$ | | |
| | Clean | PGD | AA |
| *CIFAR-100* | $\epsilon = 0.031$ | | |
| BiLAW-TRADES (ours) | 62.8 | 31.4 | 27.2 |
| AT | 57.9 | 28.9 | 24.7 |
| TRADES ($1/\lambda = 1$) | **62.4** | 25.3 | 22.2 |
| TRADES ($1/\lambda = 6$) | 56.5 | **30.9** | **26.9** |
| GAIRAT | 60.2 | 30.4 | 22.6 |
| WMMR | 56.1 | 29.7 | 25.8 |
| MAIL | **62.4** | **30.9** | 26.8 |

case, we maintain and outperform relevant methods with respect to both PGD-based and AA-based robust accuracy while achieving superior clean test accuracy. We demonstrate that when used in conjunction with TRADES, BiLAW preserves and improves robustness to AA attacks by 1.5% in contrast to TRADES, while significantly enhancing clean test accuracy by up to 3% and PGD attacks by up to 5%. We also note that parametric reweighting as opposed to the non-parametric version significantly improves robust accuracy. On CIFAR-100 (Table 3) BiLAW-TRADES out-performs all other methods. Our results demonstrate the effectiveness of using a held-out validation set to learn the sample weights compared to heuristic reweighting schemes.

In the Appendix we conduct three ablative experiments to analyze the effect of (1) the capacity of the auxiliary weighting network, (2) the fixed TRADES coefficient and (3) the input to the weighting network, on the performance of BiLAW.

## 4.2 TRAINING SAMPLE WEIGHTS

We investigate the correspondence between weights and samples, and ask the question: *what are the properties of training examples with high/low weights?* Fig. 3 provides evidence that supports our claim that samples for which the auxiliary network predicts high weights correspond to vulnerable, or difficult samples close to the decision boundary. In Fig. 3(a)-(b), we plot the distribution of weights for each class, as well the associated confusion matrix of predictions made by a robust classifier (trained with BiLAW) on adversarial samples. We note that the distribution of weights matches the distribution of misclassified adversarial examples. For example, in Fig. 3(a), samples of the 'ship' and 'automobile' class are assigned a higher number of smaller weights and they are typically classified correctly as in Fig. 3b. In contrast, birds, cats, and other animals have a higher number of samples assigned large weight and are more frequently misclassified.

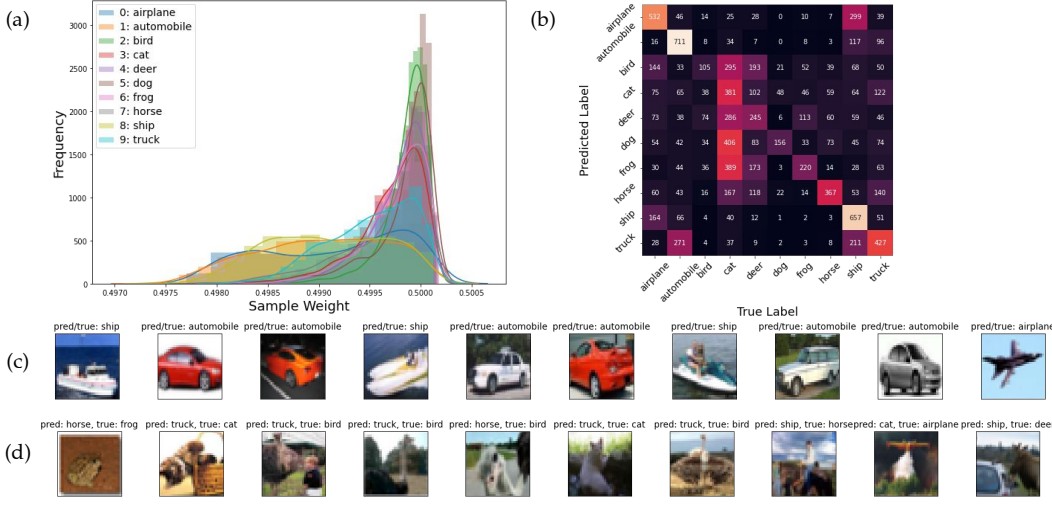

Figure 3: (**a**) Weight distribution and examples of CIFAR-10 samples. (**b**) Adversarial confusion matrix of a robust network on CIFAR-10. (**c**) "Easy" CIFAR-10 samples with low weight are correctly classified. (**d**) "Hard" CIFAR-10 samples with high weight are typically incorrectly classified.

In Fig. 3(c), we provide several examples of test samples that are assigned low weight. These images are typically very clear, involving a centered object and plain background. In Fig. 3(d), we provide a set of test samples assigned high weights. Many of these images are challenging for humans to identify, even when uncorrupted by adversarial noise. For example, the second and fifth image are pictures of cats and birds with unusual pose. The seventh, eighth, and ninth image are nearly impossible to identify due to complex backgrounds or obscured objects. Additionally, the second, third, eighth, and tenth images consist of multiple objects that could confuse the network or facilitate more effective adversarial attacks. Thus, high weights can be used to identify adversarially vulnerable samples, and automatically differentiate easy and challenging samples in existing datasets.

## 5 CONCLUSION

We have introduced **BiLAW**, a new robust training method that learns classifiers that are robust to norm-bounded adversarial attacks and is inspired by the hypothesis that reweighting via bilevel optimization offers a remedy to the issue of adversarial overfitting. We demonstrate that our method learns robust networks that out-performs competing methods, including related, geometrically-motivated techniques. Notably, **BiLAW** does not rely on complicated heuristics to assign weights, and we demonstrate the learned weights are interpretable. Future work involves improving scalability and investigating whether the auxiliary network might be used to *detect* adversarial corruptions.

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

# A APPENDIX

## A.1 DERIVATION OF META GRADIENT

In this section we derive the update rule for the parameters of the auxiliary network in Eq. 6:

$$\mu_t = \mu_{t-1} - \frac{\alpha\beta}{mn} \sum_{j=1}^{m} \left( \sum_{i=1}^{n} \left( \frac{\partial \hat{\ell}_i^{\mathrm{val}}(\tilde{\theta})}{\partial \tilde{\theta}} \bigg|_{\tilde{\theta}_t} \right)^{\top} \frac{\partial \hat{\ell}_j^{\mathrm{tr}}(\theta)}{\partial \theta} \bigg|_{\theta_{t-1}} \right) \frac{\partial w}{\partial \mu} \bigg|_{\mu_t},$$

Let

$$\hat{\mathcal{L}}_{\mathrm{tr}}(\theta_t, w) = \frac{1}{m} \sum_{j=1}^{m} w_j \hat{\ell}_j(\theta_t)$$

be the robust training loss with respect to parameters $\theta$ at time $t$ and example weight $w_j$ for the $j$-th training example. Let $\hat{\mathcal{L}}_{\mathrm{val}}(\theta_t) = \frac{1}{n} \sum_{i=1}^{n} \hat{\ell}_i(\theta_t)$ be the associated *unweighted* validation loss. Following the meta-learning framework, we to minimize this loss via gradient descent.

$$\frac{\partial \hat{\mathcal{L}}_{\mathrm{val}}(\tilde{\theta})}{\partial \mu} = \frac{1}{n} \sum_{i}^{n} \frac{\partial \hat{\ell}_i^{\mathrm{val}}(\tilde{\theta})}{\partial \mu}$$

$$= \frac{1}{n} \sum_{i}^{n} \frac{\partial \hat{\ell}_i^{\mathrm{val}}(\tilde{\theta})}{\partial \tilde{\theta}} \frac{\partial \tilde{\theta}}{\partial w} \frac{\partial w}{\partial \mu}$$

To compute $\frac{\partial \tilde{\theta}}{\partial w}$, we can apply the MAML technique and differentiate through the pseudo update (recall, $\tilde{\theta}_t = GD_{\mathrm{tr}}(\theta_{t-1}, w_{t-1}) := \theta_{t-1} - \alpha \nabla_\theta \mathcal{L}_{\mathrm{tr},t-1}(\theta_{t-1}, w)$). For example, a single gradient descent step:

$$\frac{\partial \tilde{\theta}}{\partial w} = \frac{\partial}{\partial w}(\theta_{t-1} - \alpha \nabla_\theta \mathcal{L}_{\mathrm{tr},t-1}(\theta_{t-1}, w))$$

$$= \left( \frac{\alpha}{m} \sum_{i=1}^{m} \nabla_\theta \hat{\ell}_{t-1}^{\mathrm{tr}}(\theta_{t-1}) \right)$$

So the complete update is:

$$\mu_t = \mu_{t-1} - \frac{\alpha\beta}{mn} \sum_{j=1}^{m} \left( \sum_{i=1}^{n} \frac{\partial \hat{\ell}_i^{\mathrm{val}}(\tilde{\theta})}{\partial \tilde{\theta}} \bigg|_{\tilde{\theta}_t}^{\top} \frac{\partial \hat{\ell}_j^{\mathrm{tr}}(\theta)}{\partial \theta} \bigg|_{\theta_{t-1}} \frac{\partial w}{\partial \mu} \bigg|_{\mu_t} \right)$$

## A.2 MAIN EXPERIMENTS

### A.2.1 ARCHITECTURES

We abbreviate one hidden layer fully connected network with 1024 hidden units with FC1. The tiny-CNN convolutional architecture that we use is identical to that of Wong & Kolter (2018); Croce & Hein (2020a) —consisting of two convolutional layers with 16 and 32 filters of size $4 \times 4$ and stride 2, followed by a fully connected layer with 100 hidden units. For all experiments we use training and validation batch sizes of 128 and we train all models for 100 epochs. Moreover, we use SGD with a piecewise constant learning rate schedule with initial learning rate of 0.1. The learning rate is divided by 10 at epochs 30 and 60 respectively. On all datasets (MNIST, F-MNIST, CIFAR-10, and

Table 4: Architectures for main experiments for number of classes $nc$.

| FC1 | tiny-CNN | small-CNN |
|---|---|---|
| FC(1024) | Conv(16, $4 \times 4$, 2) | small-CNN-BLOCK(64) |
| ReLU | ReLU | small-CNN-BLOCK(128) |
| FC($nc$) | Conv(32, $4 \times 4$, 2) | small-CNN-BLOCK(196) |
| | ReLU | FC(256) |
| | FC(100) | ReLU |
| | ReLU | FC($nc$) |
| | FC($nc$) | |

Table 5: Architectures for main experiments for number of classes $nc$.

| small-CNN-BLOCK($c$) |
|---|
| Conv($c$, $3 \times 3$, 1) |
| BatchNorm |
| ReLU |
| Conv($c$, $3 \times 3$, 1) |
| BatchNorm |
| ReLU |
| MaxPool($2 \times 2$) |

CIFAR-100) we restrict the input to be in the range $[0, 1]$. On the CIFAR-10 dataset, following Zhang et al. (2021), we apply random crops and random mirroring of the images as data augmentation during training. We perform adversarial training using the PGD attack of Madry et al. (2018). During training, we perform 10 iterations of the PGD attack for all datasets. During evaluation, we use 20 iterations for all datasets. Following Zhang et al. (2021), the step size is the perturbation radius divided by 4.

## A.3 ABLATION STUDY

In this section, we evaluate variations of our technique on CIFAR-10 using the WRN-32-10 architecture and $\ell_\infty$ with $\epsilon = 0.031$. First, we explore how the capacity of the auxiliary reweighting technique influences the performance of our method. Next, we evaluate our network in combination with TRADES for various values of $1/\lambda$. Finally, we demonstrate the advantage of the multi-class margin over alternative inputs mapping to the sample weights—e.g. using the class-unaware margin (Def. 1), the adversarial loss $\Delta_{\text{adv}}$, and the difference between the adversarial loss and the clean loss at a sample $\Delta_{\text{diff}}$.

Table 6: Ablation experiments: capacity of the auxiliary weight prediction network

| Capacity of $\omega$ | CIFAR10 | | |
|---|---|---|---|
| | Clean | PGD | PGD - Clean |
| 128 hidden units | 86.1 | 58.9 | 27.2 |
| 64 | 83.6 | 57.4 | 26.2 |
| $64 - 64$ | 85.8 | 58.6 | 27.1 |
| 256 | 85.7 | 57.7 | 28 |

In Table 6, we evaluate the influence of the auxiliary network capacity, i.e. the choice of $\omega$. As with training robust classifiers, the architecture of the network influences the clean-robust tradeoff, with smaller capacity networks reducing the gap between clean and robust performance, and larger networks increasing the gap.

In Table 7, we show relative robustness of our approach combined with TRADES to the choice of $1/\lambda$. In particular, we maintain an important advantage of TRADES: the ability to easily control the robustness tradeoff by controlling $1/\lambda$.

Table 7: Ablation experiments: TRADES coefficient $1/\lambda$

| $1/\lambda$ | CIFAR10 | |
|---|---|---|
| | Clean | PGD |
| $1/\lambda = 6$ | 86.1 | 58.9 |
| $1/\lambda = 1$ | 87.4 | 52.5 |
| $1/\lambda = 5$ | 86.9 | 57.6 |
| $1/\lambda = 10$ | 83.8 | 57.9 |

Table 8: Ablation experiments: $\Delta$, input to the auxiliary network

| Network input | CIFAR10 | |
|---|---|---|
| | Clean | PGD |
| multiclass margin (Def. 3) | 86.1 | 58.9 |
| margin (Def. 2) | 84.1 | 54.6 |
| $\hat{\ell}$ | 86.9 | 57.9 |
| $\hat{\ell} - \ell$ | 85.4 | 53.8 |

In Table 8, we show that the choice of input to the auxiliary neural network to predict the sample weights has a significant impact. In particular, we show the necessity of using the multi-class margin to achieve superior clean and robust test accuracy. Surprisingly, conditioning the weight on the robust loss also leads to good performance, better than the margin , and employing a learnable map for either the class-aware and class-unaware outperforms heuristic methods (e.g., WMMR and MAIL).

## A.4 CIFAR-10 EXAMPLE WEIGHTS

In Fig 4 we recover the predictions made by a small-CNN trained with BiLAW. We then use principal component analysis (PCA) to project 10-dimensional predicted class likelihoods into 2-dimensions and plot the corresponding embeddings. The color denotes the degree of the robustness of each data point. Samples which are assigned larger weight are darker. As expected, these samples associated with high weights lie close to the decision boundary and are more likely to improve robust generalization.

In Fig 5 we investigate the dynamics of predicted weights by visualizing the progression of weights predicted at margins for training samples and their adversarial variants. We observe (1) the dynamics of the weights seem to be determined largely by the learning rate of the classifier (i.e. the first adjustment to the learning rate happens around epoch 20), (2) the majority of weights predicted for *clean samples* are low (i.e. most clean samples are easy), and (3) the variance of the weight distribution is quite tight for adversarial samples.

In Fig 6 we compare weights computed via the GAIRAT and MAIL heuristics to weights predicted via BiLAW and show a positive correlation. In particular, BiLAW may be considered a generalization of the MAIL heuristic that additionally incorporates multi-class margin information. The similarity between margin-based weight estimators BiLAW and MAIL is evident, while the PGD-based GAIRAT weighting heuristic emphasizes a bimodal weight distribution.

Replicating (Fig. 3), we plot samples with small and large weight for competitive methods GAIRAT and MAIL. As with out method, samples associated with small weights appear to be "easy" and visa versa.

## A.5 MNIST WEIGHT EXAMPLES

We plot MNIST samples with small and large weight. As with CIFAR-10 (Fig. 3), samples associated with small weights appear to be "easy" in the sense that the digits are neatly written. On the other hand, digits associate with high weight are easily confused and often involve the occurrence or lack of occurrence of spaces between strokes that define certain digits (e.g. 3, 5, 0, 9, and 8).

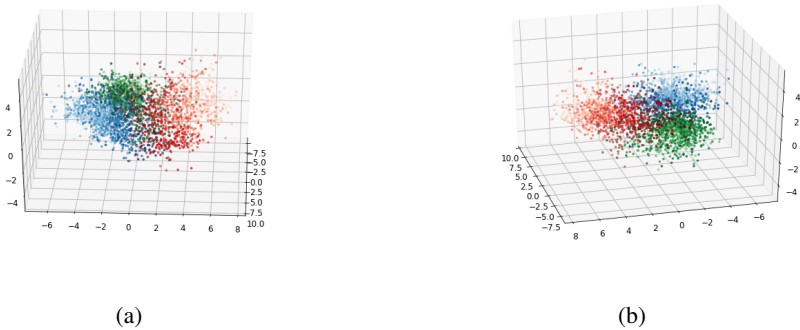

(a)                                    (b)

Figure 4: Two orientations of a 3-d plot of PCA applied to the model's likelihood predictions on training samples of three classes from the CIFAR-10 dataset (blue: car, red: plane, & green: ship). The weight of individual samples (denoted by the shade) correlates with the margin/degree of robustness.

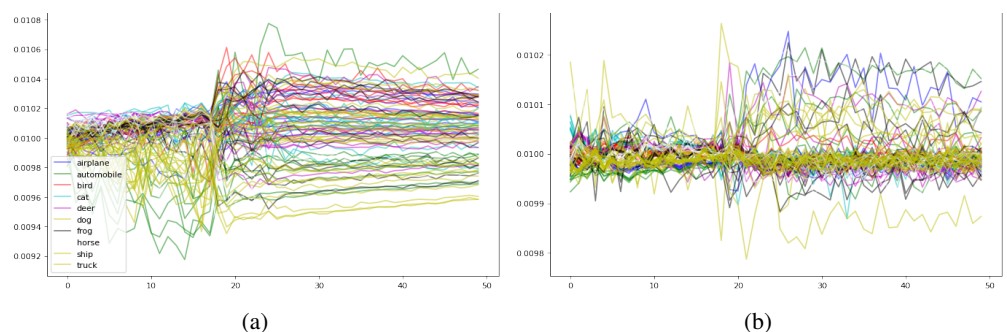

(a)                                    (b)

Figure 5: (a) Progression of weights associated with a subset of clean training samples (b) Progression of weights associated with a subset of adversarially perturbed training samples

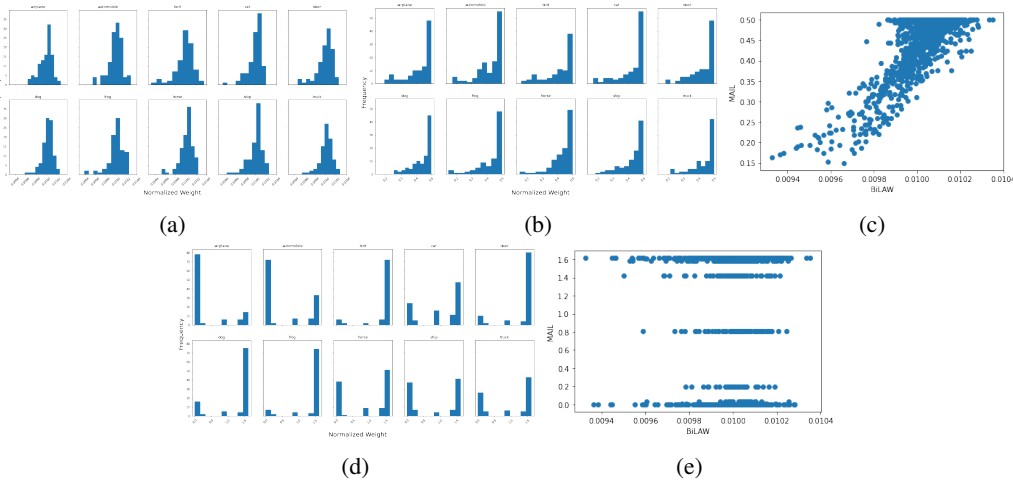

(a)                          (b)                          (c)

(d)                          (e)

Figure 6: (a) BiLAW weight distributions per-class for CIFAR-10 samples. (b) MAIL weight distributions per-class for CIFAR-10 samples. (c) Scatter plot of MAIL weight vs. BiLAW weight for a robust network. (b) GAIRAT weight distributions per-class for CIFAR-10 samples. (c) Scatter plot of GAIRAT weight vs. BiLAW weight for a robust network.

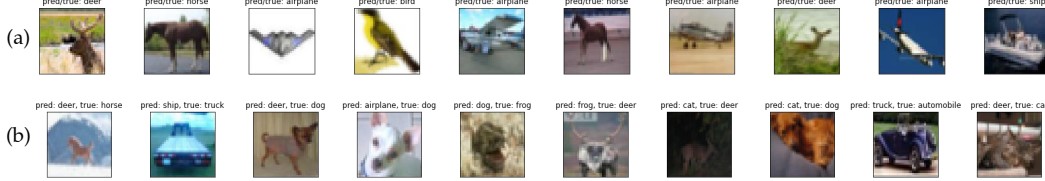

Figure 7: Examples taken from CIFAR-10 and weighted using GAIRAT (Zhang et al., 2021). (**a**) Samples with low weight. (**b**) Samples with high weight.

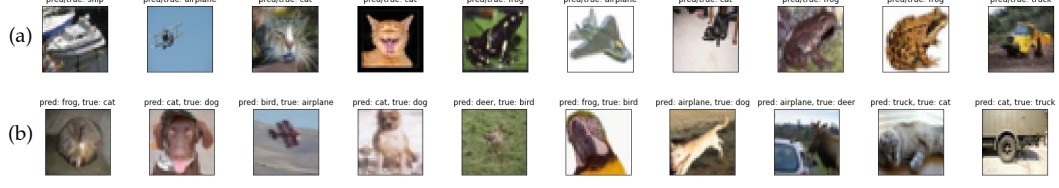

Figure 8: Examples taken from CIFAR-10 and weighted using MAIL (Wang et al., 2021). (**a**) Samples with low weight. (**b**) Samples with high weight.

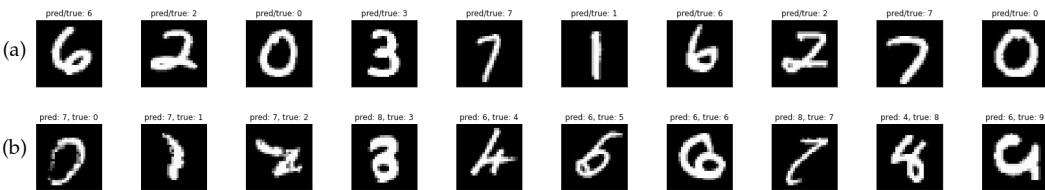

Figure 9: Examples taken from MNIST. (**a**) Samples with low weight. (**b**) Samples with high weight.

