# OpenReview forum: "Learning Sample Reweighting for Adversarial Robustness"
_ICLR.cc/2022/Conference — ICLR 2022 Submitted_

### Official Review · Reviewer_K5MU · 2021-10-25

**Correctness:** 4
**Technical Novelty And Significance:** 2
**Empirical Novelty And Significance:** 3
**Recommendation:** 8
**Confidence:** 4

**Main Review:**

The paper is well-written, clearly motivated and related work is sufficiently discussed. The comparison with the baselines is fair and the evaluation seems thorough. The improvements over the baselines are also consistent and relatively significant.

My biggest concern is that some more modern baselines may be needed in order to keep the claim in the abstract "improves both clean and robust accuracy compared to related techniques and **state-of-the-art baselines**." According to the [RobustBench leaderboard](https://robustbench.github.io/) there are many papers that achieve higher robust accuracy. Most of them use either real or synthetic auxiliary data or larger architectures, so no direct comparison is possible. However, [Adversarial Weight Perturbation Helps Robust Generalization](https://arxiv.org/abs/2004.05884) is listed as having a 3% higher robust accuracy using a "WideResNet-34-10". Is this the same or different from what the authors call "Wide-Resnet-10-32"? Ultimately, I do not think that the paper absolutely needs to compare to all baselines that achieve higher robust accuracy, because the techniques may be very different, but in my opinion the paper would benefit from clarifying where it fits in the current SOTA. For example, could the SOTA be improved by combining the technique with prior methods?

It also seems that the method's novelty is not particularly high since it simply applies a previously known technique to adversarial training. Given the good results this is not an issue though, in my opinion.

I didn't understand why the MNIST and FMNIST models were not evaluated using AutoAttack or at least Auto-PGD.

One fact that I found quite striking in the paper is that Figure 3 a) shows that the learned weights are in fact very close to one another. This makes it all the more surprising that such small differences could really have a significant impact. It would be quite interesting to see if it is really the weights that make the difference or the altered training dynamics, i.e. that the objective is changing throughout training. A simple ablation would be to retrain a model from scratch using a fixed weighting network that is taken from a previous BiLAW run. If the weights really make all the difference, then they should be unaltered under this change.

Minor comments:
- the paragraph headers on page 2 have inconsistent punctuations
- visa-versa => vice versa
- punctuations after equations are missing
- the use of $\mathcal{L}$ on page 3 seems inconsistent with its definition on page 2
- on page 5 there is a notational clash between the meaning of $\Delta_i$ and $\Delta_j$. I would recommend using a superscript for one of them.
- in step 2 on page 6 the parentheses refer to the wrong line in Alg. 1
- a boldface is missing in the 6th column of Table 2


**Summary Of The Paper:**

The paper applies ideas from meta-learning to adversarial training. They train an auxiliary network that assigns each training sample a corresponding weight which indicates how useful the training point is for robust generalization performance. They compare their robust accuracy against several baselines and demonstrate improved clean and robust accuracy as compared to their baselines.

**Summary Of The Review:**

The paper is well-motivated and has a solid evaluation showing consistent improvements on a well-studied task. Even though the authors could try to push their method's performance by additionally combining it with other baselines, in my opinion the results are significant enough to warrant a publication.

---

> ### Author Response · Authors · 2021-11-23
> **Response to Reviewer K5MU**
>
> We thank the reviewer for their positive feedback and constructive notes and for highlighting typos / notational inconsistencies.
>
> We have edited the main text to address notational and grammatical errors (including WRN-10-32 -> WRN-32-10). We have also included a new experiment in the response to all reviewers to explore the generalization of the weighting net by utilizing a pre-trained fixed weighting network to re-weight samples during training. We have also included new columns in Table 1 which highlights the performance of methods under AA attacks on the MNIST and F-MNIST datasets: BiLAW out-performs all competing reweighting methods.
>
> Below we train a robust classifier (SMALL-CNN) using BiLAW. We then save the parameters of the weighting network. Next, we initialize a new, untrained classifier. We then train this fresh classifier to minimize the weighted robust loss, where the sample weights are determined by the pretrained weighting network. Note that in this setting, the parameters of the pre-trained weighting network are fixed.
>
> | Method      | TA | PGD | AA |
> | ----------- | ----------- | ----------- | ----------- |
> | BiLAW      |  80.2     | 56.2 | 52.6  |
> | AT + pre-trained weights  |   79.4     | 55.3 | 50.9 |
>
> We observe a minor degradation in accuracy. More significantly, the performance still exceeds that of the heuristic weighting functions. This implies that the weighting network does in some sense generalize. Interesting future work might explore if this generalization extends to different classifiers or datasets (i.e. can a weighting network trained with a SMALL-CNN classifier do well be used to weight training samples for a WRN classifier, or can a weighting network trained to weight CIFAR-10 samples do well on a CIFAR-100 task?)
>
> Regarding the contribution of our method, we have added content to the main text to clarify confusion about (1.) previous work on multi-class margin vectors and (2.) state of the art heuristic adversarial defenses. In particular, as pointed out, we highlight that the majority of top-performing papers on robustbench leverage large unsupervised datasets in addition to the existing supervised dataset. Additionally, the authors of the adversarial weight perturbation paper note that the listed model on robustbench is a WRN-34-10 model, while our experiments are for a WRN-32-10 model. We plan to add additional experiments with BiLAW applied to the WRN-34-10 model. Finally, we are confident that our method “stacks” with AWP. We plan to perform this experiment.

---

> > ### Comment · Reviewer_K5MU · 2021-11-24
> > **Acknowledgement of Rebuttal**
> >
> > I acknowledge the authors' response and appreciate the additional ablation that they performed.
> > I will keep my score unchanged.
> >
> > However, I want to reiterate that, in my opinion, the claim "improves both clean and robust accuracy compared to related techniques and state-of-the-art baselines." should be weakened in the final version, unless the "stacking" with AWP is demonstrated.
> >
> > Minor comment:
> > Table 2 now says "WRN-10-32" but everywhere else it is "WRN-32-10". Please be consistent.

---

> > > ### Author Response · Authors · 2021-11-30
> > > **Addition of AWP results**
> > >
> > > Thanks. We will do a careful read through and fix all typos and inconsistencies. We intend/hope to add the AWP results to the camera-ready version. If not we will correct to “state-of-the-art reweighting baselines”.

---

### Official Review · Reviewer_3Xwo · 2021-11-01

**Correctness:** 3
**Technical Novelty And Significance:** 3
**Empirical Novelty And Significance:** 3
**Recommendation:** 6
**Confidence:** 4

**Main Review:**

Strengths (motivation, soundness, correctness, clarity):

Writing: The writing is clear and easy to understand.

Methodology: The proposed methodology is sound. The authors clearly explained how they adapted from prior works (MAML, Meta-Weight-Net, etc.). Their motivation for Def 3. and how it can be used as an input to their meta net are well explained. I also appreciate that the authors make the paper more self-contained by giving their meta gradient derivation in the appendix.

Experiments: The experimental results are good, especially under harder attacks (e.g. AutoAttack).

Weakness (novelty, improvements decoupling):
Although the proposed methodology is new in adversarial training to my best knowledge, it essentially applies idea of Meta-Weight-Net in the adversarial training setting. On the surface, this lacks some novelty. I think the paper can benefit from explaining how the performance improvement is decoupled: 1. the bilevel framework, 2. their encoding (Def 3.), say by replacing Def 3 as inputs to the meta net by simply the raw input data or the input data's activations.

Minor suggestion or typo:
- Page 3: "An immediate consequence of Eq. 1 and Def. 2.1", the hyperlink in Def. 2.1 doesn't seem to point to any definition. I cannot see Def 2.1 in the main paper either.
- Page 3: "More concretely, given a norm p and radius $\epsilon$", do the authors mean to say: ... given $L_p$ norm ...
- Page 3: I understand this notation "$L(y; x + \delta; \theta)$", but this may be more natural: $L(y; f_{\theta}(x + \delta))$, unless this is defined again in the notations section.




**Summary Of The Paper:**

The paper proposes a bilevel optimization procedure for adversarial training. While this has been done for empirical risk optimization, the paper may be the first that adapts such techniques for adversarial training. The new modification is their input encoding for the meta network: Def 3 (multi-class margin).

**Summary Of The Review:**

Overall, I think this is a nice contribution to the literature. The only drawback I can see is the lack of justification for their input encoding to the meta net, leaving the readers pondering where the improvements come from. I'd raise my score if the author can provide more justification on their input encoding (Def 3), or an ablation study.

---

> ### Author Response · Authors · 2021-11-23
> **Response to Reviewer 3Xwo**
>
> We thank the reviewer for their constructive comments and for highlighting typos. We have updated the copy to include all recommended changes.
>
> Following from previous work on reweighting based on margins, our motivation is to learn a nonlinear function of the margin, rather than pre-specify it. As we write in the paper, the multi-class margin  implicitly contains information on the correct class of the sample. Note that we have ablation studies in the appendix on the input type to the reweighting network: margin, multi-class margin, adversarial loss, difference between the adversarial loss and clean loss. The raw data and activations as inputs have high dimensionality that would increase the capacity of the reweighting network. Note also in the main text (Table 2, non-parametric weighting) we perform an ablation in which we perform a MAML bi-level optimization to update the sample weights which are a scalar, and not a parametric function as in BiLAW. Finally, we have added additional empirical evidence in updated version in the paper to demonstrate that the weights assigned by the weighting function are reasonably intuitive with respect to the margin.

---

> > ### Comment · Reviewer_3Xwo · 2021-11-27
> > **Not Directly Addressing My Concerns**
> >
> > I'm under the impression the authors don't understand my questions/concerns, since their answers appear to be out of contexts. Detailed comments below:
> >
> > `Following from previous work on reweighting based on margins, our motivation is to learn a nonlinear function of the margin, rather than pre-specify it.`
> >
> > I didn't state any function of the margin is pre-specified in the current work. Why is is raised here? My question is more about, is "output" / logit margin as input to the meta net sensible/sufficient? I interpret this as a form of inductive bias when designing the meta-net. But whether this inductive bias is superior itself can be tested.
> >
> > `As we write in the paper, the multi-class margin implicitly contains information on the correct class of the sample.`
> >
> > Yes, I agree. But my question is, why is this multi-class (logit) margin superior to other input encoding such as direct input? In my view, demonstrate this superiority against other information encoding is a good contribution. Your work appears to postulate that we should start with the output margins (of various kinds). It is not clear to me why this is the case. For example, direct margins in the input space appear to be equally if not more sensible (although it may not be computable directly). Prior works due to their non-learnable design cannot explore such potentials, but the present work can.
> >
> > `Note that we have ablation studies in the appendix on the input type to the reweighting network: margin, multi-class margin, adversarial loss, difference between the adversarial loss and clean loss.`
> >
> > Such ablation studies don't directly address my concerns. I have no concerns on the losses or multi-class margin v.s. margin, but the input information encoding for the meta net.
> >
> > `The raw data and activations as inputs have high dimensionality that would increase the capacity of the reweighting network.`
> >
> > I find this a very unsatisfying answer with little theoretical or empirical support. Why would increasing capacity of the reweghting network necessarily a problem? This requires a case-by-case examination. There are many cases where increasing capacity of neural nets can help test time performance, in both the classical supervised learning and the adversatial training framework. If the authors had such empirical studies, I'd be curious to see and encourage the authors to report them.
> >
> > `Note also in the main text (Table 2, non-parametric weighting) we perform an ablation in which we perform a MAML bi-level optimization to update the sample weights which are a scalar, and not a parametric function as in BiLAW. Finally, we have added additional empirical evidence in updated version in the paper to demonstrate that the weights assigned by the weighting function are reasonably intuitive with respect to the margin.`
> >
> > Again I find such information irrelevant and not addressing my question. The ablation study doesn't address the alternative input data encoding to the meta net. I'm not questioning whether the present method is correct, but to help uncover its potential novelty. The scalar non-parametric weighting is highly similar to bilaw, just without an outer sum multiplied by $\frac{\partial w}{\partial \mu}$, while I'm trying to see the superiority of logit based (multi) margin based encoding.
> >
> > The comparison is more pronounced in the current work, as their meta net is learnt, rather than pre-specified as in the prior works they surveyed. The meta net's input data type, how they are encoded, as well as the meta network's architectural designs can be the novel part of the paper. But I'm not seeing evidence from the authors' response.
> >
> > If my comments are not well understood, I'd appreciate that the authors can ask for clarifications. At the moment, I won't raise my score.

---

> > > ### Author Response · Authors · 2021-11-30
> > > **Thank you for the additional comments**
> > >
> > > We thank the reviewer for their clarifications and appreciate the additional comments. Our intention was to first clarify why we are using a version of margin as an input to the reweighting scheme - i.e. our initial inspiration was from previous work on margin-based sample reweighting. To further discuss:
> > >
> > > Q1. Weight predictor input and encoded information
> > >
> > > As we are inspired from margin-based work in our ablation studies we tested different inputs to the reweighting network as in our previous reply: comparing the multi-class margin to margin, loss, robust loss difference. As opposed to using the data samples themselves, these inputs are dynamic with training and reflect the current state of the network’s performance.   We agree it is of interest to explore alternative inputs as the reviewer suggests - we can add these experiments to the camera-ready version as we will be unable to update before the discussion period ends.
> > >
> > > Q2. Capacity of weighting network
> > >
> > > There might be a misunderstanding here - these are alternative inputs to the meta net as you seem to have requested, although we have not had an opportunity to include the samples directly. As explained above, we will do so. Increasing capacity of the reweighting inputs can affect timing, however we concede this might not be a major concern. We do believe that static vs dynamic inputs can affect results.
> > >
> > > Q3. Parametric and nonparametric BiLAW
> > >
> > > Regarding the similarity between nonparametric and parametric BiLAW, note that for nonparametric BiLAW, the weight associated with an individual training sample $x$ is exactly the correlation between the gradient of the loss suffered by the network at $x$ and the average over the gradients of all validation set samples. For parametric BiLAW this term appears in the gradient update for the parameters of the weighting function, and is additionally weighted by the $\frac{\partial w}{\partial \mu}$ term as you mentioned. One additional advantage of parameterizing the weighting function is that we can explore methods to reduce the computational overhead of BiLAW training- i.e. by evaluating the generalizability of a pre-trained weighting network.

---

### Official Review · Reviewer_E9Gz · 2021-11-02

**Correctness:** 3
**Technical Novelty And Significance:** 2
**Empirical Novelty And Significance:** 3
**Recommendation:** 6
**Confidence:** 4

**Main Review:**

Though the concept of reweighing training samples based on margins to improve robustness is not new, the paper presents novel approaches to reweighing by a) using multi-class margins and b) meta-learning how to reweight the samples from the class margins. The experiments show that BiLAW generally outperforms the current baselines. The paper is generally easy to follow.

The main weaknesses of the paper are the lack of 1) theoretical discussion/justification of why the use of proposed multi-class margin is better and 2) ablation studies to separately test that the two proposed components a) multi-class margin and b) meta-learned margin-to-weight mapping indeed improve the performance as claimed.

Comments & Questions:
What is the computational cost of BiLAW versus other reweighting baselines?
What could explain BiLAW underperforming some of the baselines in some settings?

Typo:
Testcases > test cases


**Summary Of The Paper:**

The paper proposes a novel approach (BiLAW) to reweigh training samples with the aim of improving models’ adversarial robustness. Different from previous reweighing methods, BiLAW uses multi-class margins and a trained neural network that learns to map the class margins to sample weights. The author claims that these two components improve sample reweighting to bring better robustness and clean accuracy. A meta-learning algorithm (MAML) is used to train the reweighing model. Experiments are conducted in the MNIST, CIFAR-10 and -100 datasets to show that BiLAW outperforms previous robustness and reweighting baselines.

**Summary Of The Review:**

Despite the weaknesses and the limited novelty from previous reweighting algorithms, I am more inclined to accept given the improved empirical results.

---

> ### Author Response · Authors · 2021-11-23
> **Response to Reviewer E9Gz**
>
> We thank the reviewer for their positive feedback and constructive notes. To respond to the questions:
>
> Q1 - Multi-class margin justification
>
> We highlight that multi-class margins have been investigated before in the context of boosting and kernel learning [1-3]. We will add a review of these papers to the revised paper. In particular, [1,2] capture the recent formulations of the robust margin that we directly compare to. However, there are still differences with our method, and in particular, our formulation contains strictly more information compared to [1,2]. Additionally, as far as we are aware, we are the first to introduce the concept of the multi-class margin vector in the context of neural network training and adversarial robustness.
>
> [1] Zou, Zhu, Hastie, The Margin Vector, Admissible Loss and Multi-class Margin-based Classifiers, 2005
>
> [2] Saberian, Vasconcelos, Multiclass Boosting: Margins, Codewords, Losses, and Algorithms, JMLR’19
>
> [3] Cortes, Mohri, Rostamizadeh, Multi-Class Classification with Maximum Margin Multiple Kernel, ICML’13
>
>
> Q2 - Ablation studies
>
> The multi-class margin implicitly contains information on the correct class of the sample. Note that we have ablation studies in the appendix on the input type to the reweighting network: margin, multi-class margin, adversarial loss, difference between the adversarial loss and clean loss. Note also in the main text (Table 2, non-parametric weighting) we perform an ablation in which we perform a MAML optimization to update the weights which are a scalar, and not a parametric function.
>
> Q3 - Relative performance of BiLAW
>
> It is important to note that there is always a tradeoff between clean and adversarial test performance. Therefore when comparing the methods we take both into account, for example as evident below (and in the ref you provided) improving the clean test error for TRADES, as you mentioned, leads to worse robust error. More concretely, we would like to clarify the following:
> In comparison to the relevant sample reweighting methods and also standard AT, we have better clean and robust accuracy for all attacks.
> Below, we compare the performance of BiLAW-TRADES to TRADES  on CIFAR-10 using the WRN-32-10 network for two values of $1/\lambda$ (limited due to time constraints), where the results are organized in two categories (similar clean test error and similar PGD-based robust test error). We note several things:
>
> (1) when TRADES ($1/\lambda= 1$) and BiLAW exhibit similar clean test accuracy, we considerably outperform TRADES with respect to test-set robustness to both PGD and AA-based attacks.
>
> | Method      | TA | PGD | AA |
> | ----------- | ----------- | ----------- | ----------- |
> | TRADES, $1/\lambda=1$      |  87.2     | 48.1 | 45.5  |
> | BiLAW  |   86.1     | 58.9 | 53.6 |
>
> (2) When TRADES ($1/\lambda= 6$) and BiLAW exhibit similar AA robustness, we outperform TRADES with respect to clean test-set accuracy and PGD-based robustness.
>
> | Method      | TA | PGD | AA |
> | ----------- | ----------- | ----------- | ----------- |
> | TRADES, $1/\lambda=6$      |  83.1     | 53.9 | 52.1 |
> | BiLAW  |    86.1    | 58.9 | 53.6 |

---

> > ### Comment · Reviewer_E9Gz · 2021-11-28
> > **Thank you for the Response**
> >
> > I thank and acknowledge the authors for their response. While some of the issues are addressed, the main weakness of the paper such as theoretical justification and incremental novelty remains. I am, hence, retaining my original score.

---

### Official Review · Reviewer_tCDo · 2021-11-02

**Correctness:** 4
**Technical Novelty And Significance:** 3
**Empirical Novelty And Significance:** 3
**Recommendation:** 8
**Confidence:** 4

**Main Review:**

Strengths:
1. The technique of learning the sample weights during training instead of simply using a function proportional to class margins is novel and interesting. Further, similar bi-level optimization has been used for other tasks (MAML, GAN training) for sample reweighing with a fair amount of success.
2. The claims in the paper are well supported with experiments and comparisons are adequate.
3. The authors also demonstrate the need for a new definition of multi-class margin through a very interesting experiment ranking the adversarial logit index.

Weaknesses/Questions:
1. How will the defense fair against an adaptive attack? In this case, if the attacker knows the relative average weights of the various classes, the attack algorithm may be able to compensate for that.
2. Sec 4.2 shows the distribution of high and low weighed training samples for BiLAW. It would be great to see similar plots for GAIRAT and MAIL to see if the weights are a property of the dataset or that of the training method.
3. The ablation studies in the appendix should be in the main paper in order to demonstrate the advantages of BiLAW over other methods such as TRADES. The paper would benefit from a more involved discussion of the ablation instead of Sec 4.2.
4. I am also curious about training times for BiLAW models. Is there a significant difference in the number of epochs required in comparison with TRADES?

**Summary Of The Paper:**

The authors propose a bi-level adversarial training mechanism to learn sample weights with the goal of training more robust neural networks.  In comparison with related work which derive sample weights using various definitions of class margins, the authors parametrize the weights using a neural network. They show the efficacy of their approach by testing the robustness of their model against a variety of attacks.

**Summary Of The Review:**

Overall, the paper is well written. I especially like the way the authors have motivated BiLAW, and appreciate the detailed comparisons with related work. The improvements while not significant, do merit publication and further discussion.The experiments are also comprehensive and support all claims. I am therefore inclined to recommending an accept.

---

> ### Author Response · Authors · 2021-11-23
> **Response to Reviewer tCDo**
>
> Thank you for your positive feedback!  To respond to the questions:
>
> Q1 - Adaptive attacks
>
> Exploring weight-aware attacks is an interesting idea, but out of the scope of this work. We have provided further discussion in the response to all reviewers.
>
> Q2 - plots of weights
>
> Thank you for your suggestion - we have added plots to the appendix: histograms of weights for GAIRAT / MAIL as well as examples of samples of low/high weights for both methods for the CIFAR-10 dataset.
>
> Q3 - Training time
>
> For all models, including TRADES and BiLAW / BiLAW-TRADES, we train for 100 epochs. It is certainly possible that with additional epochs and a more refined tuning schedule that we may achieve better results. Furthermore, we hypothesize that our method may improve when combined with more techniques orthogonal to TRADES (e.g. Adversarial Weight Perturbation).

---

> ### Comment · Reviewer_tCDo · 2021-11-24
> **Thanks for the response!**
>
> I thank the authors for the detailed response and clarifying my questions. While I still believe a discussion of weight-aware attacks is warranted here, the paper, as it is, does test against standard attacks. I stand by my earlier review and still propose the acceptance of this work.

---

### Official Review · Reviewer_oJtx · 2021-11-09

**Correctness:** 3
**Technical Novelty And Significance:** 2
**Empirical Novelty And Significance:** 2
**Recommendation:** 3
**Confidence:** 4

**Main Review:**

Strengths:
1. It is interesting to combine MAML with adversarial training and formulate a bi-level reweighing adversarial training framework.
2. The definition of multi-class margin is intuitive but enlightening for rethinking the boundary of samples.
3. The structure of the article is relatively organized.

Weaknesses:
1. My biggest concern is that, from Table 2 and 3, using BiLAW alone is not competitive with others when facing more reliable attack methods like AutoAttack, especially TRADES. So when used in combination with TRADES, the gain is more likely to be brought by TRADES rather than BiLAW itself. Combined with the higher robust accuracy against PGD, it suggests that the boundary of the model trained by BiLAW may not significantly change. BiLAW is more likely to make the area near some hard samples (for PGD and its variants only) much sharper, which is virtually harmful to construct a certified robust model.
2. It seems that the uncertainty of the source of performance improvement. It is hard to say whether such robust performance comes from the reweighting strategy or just the increase in parameters or the structural change brought by the auxiliary network. Although this network will not be activated in the inferring stage, the inaccessibility of the auxiliary network to attackers may result in a performance boost.
3. Since the authors add an auxiliary network to train the model. It is natural to care about the running time for BiLAW in comparison with other methods.

**Summary Of The Paper:**

The authors propose a bi-level adversarial training method to reweight the importance of samples in each mini-batch, aiming at building a more robust DNN model. The authors borrow the idea from meta-learning and design an auxiliary network to learn such weights. The results show their approach can improve robustness against certain attacks.

**Summary Of The Review:**

Overall, considering the weaknesses, not good enough experimental results and the limited novelty, I am inclined to reject the paper unless the authors can provide more interesting discovery and explain a compelling reason why it cannot perform well itself.

---

> ### Author Response · Authors · 2021-11-23
> **Response to Reviewer oJtx**
>
> We thank the reviewer very much for their comments and suggestions.
>
> Q1 & Q2 - Tables 2 and 3
>
> For both tables, the left column denotes the defense technique. We highlight that our framework yields classifiers which match or outperform relevant methods with respect to clean and robust errors. In particular, BiLAW + TRADES beats TRADES in both tables. Furthermore, BiLAW without TRADES has comparative performance to TRADES on AA and out-performs TRADES on clean and PPGD. BiLAW with TRADES improves on clean accuracy by 3% and on PGD by 5%, so we can conclude these gains are due to BiLAW.
>
> It is important to note that there is always a tradeoff between clean and adversarial test performance. Therefore when comparing the methods we take both into account, for example as evident below (and in the ref you provided) improving the clean test error for TRADES, as you mentioned, leads to worse robust error. More concretely, we would like to clarify the following:
> In comparison to the relevant sample reweighting methods and also standard AT, we have better clean and robust accuracy for all attacks.
> Below, we compare the performance of BiLAW-TRADES to TRADES  on CIFAR-10 using the WRN-32-10 network for two values of $1/\lambda$ (limited due to time constraints), where the results are organized in two categories (similar clean test error and similar PGD-based robust test error). We note several things:
>
> (1) when TRADES ($1/\lambda= 1$) and BiLAW exhibit similar clean test accuracy, we considerably outperform TRADES with respect to test-set robustness to both PGD and AA-based attacks.
>
> | Method      | TA | PGD | AA |
> | ----------- | ----------- | ----------- | ----------- |
> | TRADES, $1/\lambda=1$      |  87.2     | 48.1 | 45.5  |
> | BiLAW  |   86.1     | 58.9 | 53.6 |
>
> (2) When TRADES ($1/\lambda= 6$) and BiLAW exhibit similar AA robustness, we outperform TRADES with respect to clean test-set accuracy and PGD-based robustness.
>
> | Method      | TA | PGD | AA |
> | ----------- | ----------- | ----------- | ----------- |
> | TRADES, $1/\lambda=6$      |  83.1     | 53.9 | 52.1 |
> | BiLAW  |    86.1    | 58.9 | 53.6 |
>
> Q2 - Adaptive attacks
>
> Exploring weight-aware attacks is an interesting idea, but out of the scope of this work. We have provided further discussion in the response to all reviewers. In particular, the reweighting strategy is inaccessible to the attackers for methods that employ reweighting since reweighting is only performed during training, and is independent of the model architecture/parameters. Furthermore, there is no increase in the number of parameters of the classifier- the capacity of the classifier is unchanged at test-time, and the network parameters are accessible to the attacker.

---

### Official Review · Reviewer_fjzH · 2021-12-05

**Correctness:** 3
**Technical Novelty And Significance:** 2
**Empirical Novelty And Significance:** 2
**Recommendation:** 3
**Confidence:** 4

**Main Review:**


Strengths:
- As for as I know, it is the first learning-based re-weighting strategy for adversarial robustness, compared to previous heuristic methods like GAIRAT.
- The paper is well-written and easy to follow. The proposed approach is methodologically sound.
- The evaluation includes strong adversaries, like AutoAttack, and they show their reweighing does not suffer from performance degeneracy like GAIRAT.

**Major Concerns**:

**1. Novelty.** The proposed approach is a direct adaptation of the classical MAML algorithm to adversarial training, which is of limited technical novelty and hardly meets the bar of this venue. Deeper investigation of the reweighting mechanisms should be discussed, for example, the effect on the multi-class margin by the proposed reweighting, should be demonstrated.

**2. Evaluation.** Unlike previous approach, the proposed BiLAW relies an additional reweighting module in the training stage. Therefore, in the (fully transparent) white-box setting, I believe this additional module should be included in the evaluation stage for crafting adaptive attack. I have read the authors’ response to other reviewers, while I cannot agree with that evaluating adaptive attack is “beyond the scope of this paper”. Instead, because the proposed method distinguishes itself as using a learned reweighting, the authors cannot use other methods for an excuse, and the evaluation of adaptive attack on the learned reweighting module is **within the scope of this work**. Even if the authors argue that the reweighting module is not included in the test stage, we can still use an independently learned reweighting head for adaptive attack. To wrap up, I believe this work would not be complete without evaluating adaptive attack, and I strongly suggest that the authors should do so in the future.

**3. Performance.** The improvement is still marginal under AA, and it relies on a combination with TRADES to obtain good accuracy, and lacks comparison to SOTA baselines like AWP. MNIST/F-MNIST is too old for the current research, and small CNNs are also unnecessary for CIFAR-10. On CIFAR-100, the result of BiLAW is **not shown**. I encourage the authors to be honest with the results even if it is relatively lower.

Minor concerns:

The authors spend too much space on preliminaries and lack enough discussion on introducing and justifying their own method. For example, the technical discussions of multi-class margin (Section 3.1) is unnecessarily complex, as the authors have not shown any rigorous connection between the theory and their proposed method. Also, in Step 2 (Section 3.2), the update rule is also unnecessarily complex, as the sum term could be simply replaced by the average loss.


**Summary Of The Paper:**

This paper develops a meta-learning approach for re-weighting samples for better adversarial robustness. In particular, they parameterize the weights using an additional module and learn it with the MAML objective. Through a comparison with previous work, they show that the proposed method improves model robustness on benchmark datasets.


**Summary Of The Review:**

Based on the limitations above, I think that this paper, although having technical contributions to the community, is still not ready for publication due to the lack of fair evaluation. Therefore, I recommend rejection. I would suggest the authors address my concerns (as well as those from other reviewers) with additional experiments and submit it to a future venue.

---

> ### Author Response · Authors · 2021-12-08
> **Response to Reviewer fjzH**
>
> We thank the reviewer for their constructive and helpful comments. To clarify adaptive attacks, we discuss two cases: when the weights are treated as constants and for the case when the weights are treated as a function of the classifier and labels. We perform a preliminary experiment to explore the feasibility of a weight-aware variant of PGD.
>
>
> 1 - Novelty
>
> It is true that our method shares similarity with the MAML algorithm. However, Bi-level optimization is a very general framework and there are few techniques that may be employed to solve such problems efficiently, especially in the context of neural networks and adversarial robustness. Alternating gradient descent is an effective strategy to apply for our use case, where our primary contribution is the formulation of learning a parametric reweighting function and implementation of the multi-class margin. We agree that our empirical results motivate deeper investigation into the reweighting mechanisms. We will add additional experiments in the final version.
>
> 2 a. - weight-adaptive attacks ($w_i$ constant)
>
> If the $w_i$ are considered a constant, even if the attacker has full knowledge of the weighting network, that the attack will not change (i.e. the adversarial perturbation will be the same regardless of knowledge about the weighting network). Here is the reason:
> consider an untargeted attack
> $$
> \max_{\delta} \mathcal{L}(f(x_0+\delta,\theta), y_0) \quad \textrm{(1)}
> $$
> where $x_0$ is original image, $y_0$ is the true label of $x_0$, $\delta$ is the adversarial perturbation to be solved subject to the constraints $||\delta||_p \leq \epsilon$, $x_0 + \delta \in [0,1]^n$, and $\theta$ are the classifier parameters. An adaptive attack would involve adding the weight (known to the attacker, not the classifier) of the test sample $x_0$ (say denoted as $w_0$), which means we solve:
> $$
> \max_\delta w_0*\mathcal{L}(f(x_0+\delta, \theta), y_0) \quad \textrm{(2)}
> $$
> The solution of (1) and (2) will be the same as long as the weight is positive (which is guaranteed via a normalization layer).
>
> 2 b. - weight-adaptive attacks ($w_i$ function)
>
> If we consider each $w_i$ as a function of the classifier, an attacker could indeed perform gradient ascent on the loss suffered by the classifier at an input $x_i$.
> In particular, the gradient of the perturbation would be decomposed into the sum of two parts:
> 1. One part would just be the typical adversarial direction scaled by the predicted weight (a function of the margin): $w(y_i - f(x_i + \delta)) * \frac{\partial}{\partial\delta}(\mathcal{L}(y_i, f(x_i + \delta)))$
> 2. One part would involve gradient of weighting network wrt delta: $\mathcal{L}(y_i, f(x_i + \delta)) * \frac{\partial}{\partial\delta} w(y_i - f(x_i + \delta))$
>
> This implies that the solution will be different for (1) $\max \mathcal{L}(y_i, f(x_i + \delta))$ and (2) $\max w_i*\mathcal{L}(y_i, f(x_i + \delta))$. This means that the "adaptive" attack will only give an equal or worse answer to (1) if solved to optimal. The reason is that, (1) is the true formulation of the adversarial perturbation, while (2) is not. This means that the "adaptive" attack would only be weaker than the original attack if $w_i$ is dependent on $\delta$ in theory. As evident by the new experiment result, the adaptive attack result is slightly worse than the standard attack result. We evaluate the adaptive attack on a pre-trained SMALL-CNN network to validate this. We evaluate an attacker that has full knowledge of the weighting network and an attacker which only has access to a pre-trained weighting network:
>
> | Method      | clean test accuracy | PGD accuracy |
> | ----------- | ----------- | ----------- |
> | standard pgd (weight-unaware)  |   80.2      | 56.2  |
> | adaptive pgd (oracle weight)     |  –     |  56.8  |
> | adaptive pgd (pre-trained weight)     |  –     | 57.3 |
>
> In the final draft of the paper, we will perform a more extensive analysis, but these preliminary results imply that even oracle knowledge of the weighting network does not help an attacker.
>
> 3 - Performance
>
> Note that several recent robustness papers do provide experiments on MNIST/F-MNIST (e.g. [1,2,3,4]]) on a variety of small and large architectures. I.e. to show the efficacy of BiLAW-based approaches with respect to both clean/adversarial accuracy (adversarial overfitting) on a variety of problem settings and different architectures. Although CIFAR-100 is rarer in the literature, we thank the reviewer for their recommendation- we will add BiLAW to Table 3 in the final draft.
>
> [1] Croce et al., Provable Robustness of ReLU networks via Maximization of Linear Regions, AISTATS'19
>
> [2] Zhang et al., Theoretically Principled Trade-off between Robustness and Accuracy, ICML'19
>
> [3] Croce & Hein, Provable robustness against all adversarial $\ell_p$-perturbations for $p>1$, ICLR'20
>
> [4] Croce & Hein, Reliable Evaluation of Adversarial Robustness with an Ensemble of Diverse Parameter-free Attacks, ICML'20

---

### Author Response · Authors · 2021-11-23
**Summary of responses**

We thank the reviewers for their constructive and helpful comments. In the revision we marked corrections and new text in blue.

In response to general questions:


1 - Runtime

Regarding the computational cost of our algorithm, we note that compared to standard adversarial training, the primary overhead of our algorithm consists of computing the pseudo-updates and  meta-gradients (steps 1 and 2 in section 3.2). In other words, compared to T-step PGD-based adversarial training, which requires O(T+1) backward passes through the classifier for each update, BiLAW requires O(2*T) backward passes through the classifier to compute the adversarial perturbations for training and validation samples, 1 backward pass through the classifier to compute the pseudo-update, and a backward pass through the classifier and weighting network to compute the update for the weighting network. Note that the capacity of the weighting network is typically negligible compared to that of the classifier (e.g. for the main experiments we use a single-hidden layer network with 128 neurons) Considering the better performance, this extra cost should be affordable. We additionally highlight that the auxiliary network does not influence the prediction run-time or attack runtime.


2 - Additional motivation for the multi-class margin

We highlight that multi-class margins have been investigated before in the context of boosting and kernel learning [1-3]. We briefly review these papers in the revised paper. In particular, [1,2] capture the recent formulations of the robust margin that we directly compare to. However, there are still differences with our method, and in particular, our formulation contains strictly more information compared to [1,2]. Additionally, as far as we are aware, we are the first to introduce the concept of the multi-class margin vector in the context of neural network training and adversarial robustness.

[1] Zou, Zhu, Hastie, The Margin Vector, Admissible Loss and Multi-class Margin-based Classifiers, 2005

[2] Saberian, Vasconcelos, Multiclass Boosting: Margins, Codewords, Losses, and Algorithms, JMLR’19

[3] Cortes, Mohri, Rostamizadeh, Multi-Class Classification with Maximum Margin Multiple Kernel, ICML’13


3 - Clarification of ablative experiments and weight-adaptive attacks

We clarify the ablative experiments we have included in the main text. In the appendix we conduct 3 experiments to answer the questions: (1. Table 6) How does the capacity (architecture) of the weighting network influence clean and robust classification performance? We evaluate various architectures to model the weights. (2. Table 7) How does the fixed TRADES coefficient influence clean and robust classification performance (i.e. to what degree can we still control the clean-robust accuracy tradeoff?) We evaluate various settings of the $1/\lambda$ term. (3. Table 8) How does the input to the weighting network influence the performance of BiLAW? We evaluate inputs encoding various levels of information.

Regarding weight-adaptive attacks, the auxiliary network requires label information to estimate the weight. At test time, the label is not available. If the auxiliary network and label is available to the attacker, it might be interesting to explore: (1.) attackers which sample test-data non-uniformly (e.g. according to the weight predictions) (2.) iterative attacks which are weighted somehow according to the margin/weight prediction.


4 - New experiments in appendix

We have uploaded a new version of the paper. Notably, we have included 3 new figures recommended by reviewer tCDo: plots of CIFAR-10 samples associated with high weights as computed by GAIRAT and MAIL for a robust network.

Additionally, we have provided further evidence for the interpretability of BiLAW weights computed from multi-class margins. We do this by applying PCA to the output logits of a robust network evaluated on CIFAR-10 samples taken from three classes and shading the samples according to the associated weight.

Below, we summarize our reviewer responses below and provide detailed responses in the reply to each reviewer.

For Reviewer oJtx:
- We have clarified the format of Table 2 and Table 3.
- We have provided a more detailed analysis and comparison between our method and TRADES

For Reviewer tCDo:
- We have discussed the inclusion of new ablative experiments covering weight dynamics and histograms of alternative methods

For Reviewer E9Gz: We have
- We have discussed the inclusion of new ablative experiments and computational cost of the method.

For Reviewer 3Xwo:
- We have made the recommended edits to the paper and have clarified the ablative experiments.

For Reviewer K5MU:
- We have included new ablative experiments and results for AA on the MNIST and FMNIST datasets
- We have included a new experiment exploring the generalization of the weighting network - i.e. we utilize a pre-trained weighting network during training.

---

### Decision · Program_Chairs · 2022-01-20

**Decision:**

Reject

**Comment:**

The paper introduces a meta-learning approach for re-weighting samples for better adversarial robustness. Specifically, they parameterize the weights using an additional module and learn it with the MAML objective. I have read the paper and reviews carefully by myself and found that the paper has several weaknesses that are not well addressed in the rebuttal.

1) limited novelty. The proposed approach is a direct adaptation of the classical MAML algorithm to adversarial training, which is of limited technical novelty as pointed by serveral reviewers.

2) Adaptive attack experiments are incomplete. The proposed BiLAW relies an additional reweighting module in the training stage, though it will not be used in the test stage. But we can still use an independently learned reweighting head for adaptive attack, which is should be considered in the white-box attacks. We do not want to see the new proposed defense will be defeated by other attacks quickly.

3) The true performance for BiLAW is problematic. Table 1 on MNIST is not representative for current development in adversarial community. On Table 2, comparing BiLAW with TRADES, for AA (0.031), 45.3% vs 51.7% on small-CNN and 51.4% vs 52.1% on WRN-32-10. The performance of BiLAW is lower than TRADES, while when combine the two together, the results is very natural higher than TRADES and BiLAW but we are sure which component benefit the gains. For example, we can say TRADES benefits BiLAW because BiLAW+TRADES (52.6%) is much higher than BiLAW (45.3%). Also, the author did not show the results of single BiLAW on CIFAR-100. Considering the around two times running time, this performance is not acceptable in adversarial training methods.

Due to the above reasons, I cannot recommend acceptance in the current verison to ICLR.